# Genetic Variants Underlying Plasticity in Natural Populations of Spadefoot Toads: Environmental Assessment versus Phenotypic Response

**DOI:** 10.3390/genes15050611

**Published:** 2024-05-11

**Authors:** Andrew J. Isdaner, Nicholas A. Levis, Ian M. Ehrenreich, David W. Pfennig

**Affiliations:** 1Department of Biology, CB#3280, University of North Carolina, Chapel Hill, NC 27599, USA; aisdaner@live.unc.edu (A.J.I.); nicholasalevis@gmail.com (N.A.L.); 2Department of Biology, Indiana University, Bloomington, IN 47405, USA; 3Molecular and Computational Biology Section, Department of Biological Sciences, University of Southern California, Los Angeles, CA 90089, USA; ian.ehrenreich@usc.edu

**Keywords:** developmental plasticity, genomics, GWAS, phenotypic plasticity, resource polyphenism, spadefoot

## Abstract

Many organisms facultatively produce different phenotypes depending on their environment, yet relatively little is known about the genetic bases of such plasticity in natural populations. In this study, we describe the genetic variation underlying an extreme form of plasticity––resource polyphenism––in Mexican spadefoot toad tadpoles, *Spea multiplicata*. Depending on their environment, these tadpoles develop into one of two drastically different forms: a carnivore morph or an omnivore morph. We collected both morphs from two ponds that differed in which morph had an adaptive advantage and performed genome-wide association studies of phenotype (carnivore vs. omnivore) and adaptive plasticity (adaptive vs. maladaptive environmental assessment). We identified four quantitative trait loci associated with phenotype and nine with adaptive plasticity, two of which exhibited signatures of minor allele dominance and two of which (one phenotype locus and one adaptive plasticity locus) did not occur as minor allele homozygotes. Investigations into the genetics of plastic traits in natural populations promise to provide novel insights into how such complex, adaptive traits arise and evolve.

## 1. Introduction

Phenotypic plasticity—the ability of an organism to alter its phenotype in response to changes in the environment—is ubiquitous [1,2,3]. This ability to shift aspects of the phenotype, reversibly or irreversibly, is an essential feature of an organism’s ecological interactions and evolution [2]. Plasticity is thought to be involved in mediating processes as diverse as inter- and intraspecific competition [4,5,6,7,8], the evolution of novel traits [9,10,11], speciation and adaptive radiation [12,13,14,15,16,17], and even major transitions in the evolution of individuality and sociality [18].

Although the evolutionary consequences of plasticity have garnered significant interest [2,13,19], less is known about the genetic mechanisms underlying plastic phenotypes [20,21]. Illuminating the genetic basis of plasticity is essential for elucidating the origins and evolution of phenotypic plasticity. Although it is now widely accepted that phenotypic plasticity is underlain by a complex, polygenic genetic architecture [22], we know little about the relative importance of, or the distinction between, the genetic variants involved in environmental assessment and those involved in producing the phenotypes induced by the environmental cues [20,23,24,25,26,27,28].

Consider that two key processes are necessary for an organism to generate a plastic response to the environment [13,29,30]. First, the organism must assess a cue from the environment [20]. This cue may take many forms, including chemical compounds secreted by a predator, prey item, mate, or competitor [31,32]; mechanical interactions with something in the environment, such as the chewing of food [33] or direct contact between individuals [34]; molecules absorbed during digestion [35]; visual cues, such as coloration or light [36,37,38]; and climatic factors, such as temperature [39,40] and rainfall [41]. Indeed, an organism may use more than one of these cues to assess the environment [42,43,44]. Second, the organism must successfully convert that cue into an expressed phenotype [20]. Additionally, if the plastic response is adaptive, the cue assessment must be accurate, and the generated phenotype must be well-suited to the environment with which the cue is associated.

Disentangling the environmental assessment and developmental response aspects of plasticity is necessary to understand the genetic bases of plasticity and the relationship between loci involved in both of these aspects. In time, characterizing the genetic bases of plasticity will help us to understand the types of loci that may be targeted by selection (i.e., upstream or downstream processes [20,45]) in the evolution of both the assessment and response aspects of adaptive plasticity. For example, because they are likely to have more pleiotropic effects, upstream elements have less capacity to undergo evolutionary change than downstream components in the evolution of phenotypic plasticity [46,47]. However, to evaluate such hypotheses, we must first determine which loci are associated with these complex phenotypes and establish the degree to which they are under independent selection.

A significant hurdle to overcome is that an organism that has undergone a plastic developmental process has already acquired the resultant phenotype, which makes it difficult to tease apart the genetic variants associated with environmental assessment versus the developmental response. To help address this challenge, one can examine individuals expressing adaptive and maladaptive outcomes in different selective environments [48]. By examining individuals that express both the adaptive and maladaptive outcomes in a given set of environments, one can isolate the effects of the environmental assessment from those of the developmental response. This is possible because genetic variants associated with the developmental response should exhibit an association with the produced phenotype, but variants associated with the assessment aspect of plasticity can be expected to show an association with having the adaptive phenotype favored under each environmental condition. In other words, the produced phenotypes will differ between environments such that individuals with the genotype associated with accurate environmental assessment will produce the adaptive phenotype favored by the environmental cues it experiences.

Herein, we undertake such an approach by leveraging a natural experiment that decoupled the assessment and production components of plasticity in Mexican spadefoot toads, *Spea multiplicata*. The tadpoles of this species exhibit an extreme form of plasticity known as polyphenism (*sensu* [49]), wherein two or more distinct environmentally induced phenotypes can occur in a single population. Specifically, *S. multiplicata* tadpoles may develop into one of two distinct resource-use phenotypes [50]: a more typical ‘omnivore’ morph or a novel ‘carnivore’ morph [51] (Figure 1a). Importantly, and as described in greater detail below, these two morphs are differentially favored when faced with different environmental conditions [48,50,51,52,53] (Figure 1b).

To characterize aspects of the genetic bases of resource polyphenism in *S. multiplicata*, we conducted a genome-wide association study to identify loci associated with morph production and the ability to assess (and, therefore, adaptively respond to) the environment accurately. For this study, we utilized samples that were gathered for a previously published study [30]. The present study extended the previous study by generating an improved pseudogenome assembly and determining the genetic architecture of developmental responses and environmental assessment while accounting for the relatedness among the samples. In this study, we additionally identified candidate genes containing, or in close proximity to, these loci. These results emphasize both the potential complexity of plasticity’s genetic architecture and the challenges involved in unraveling the genetic bases of the assessment and response aspects of plasticity.

## 2. Materials and Methods

### 2.1. Study System

As introduced above, *S. multiplicata* tadpoles may develop as either of two distinct resource-use phenotypes [50]: a default ‘omnivore’ morph that primarily eats detritus and microorganisms or an alternative ‘carnivore’ morph that develops when the tadpoles are exposed to and consume live prey (e.g., fairy shrimp and other tadpoles early in development [51]). Although omnivores are characterized by small jaw muscles, smooth mouthparts, a large number of denticle rows, and a long gut (traits that are well-suited to their diet [51,54]), carnivores exhibit large jaw muscles, jagged mouthparts, and a short gut that are adaptive for the handling and digestion of larger, mobile prey items [52] (Figure 1a).

The number of carnivores that develop in each population, and how extreme the carnivore-like morphology of those individuals is, varies among populations and even between sibships within a single population [55,56,57,58]. These differences suggest that there is heritable variation underlying the frequency and degree of carnivore production in *S. multiplicata* tadpoles. Another consequence of this variation is that tadpoles of both morphs are present in many ponds, especially those ponds that favor the carnivore form [50,59].

Generally, each morph is favored under different environmental conditions. Carnivores are favored in shrimp-rich, short-duration ponds because they develop faster [48,50]. By contrast, omnivores are favored in shrimp-poor, longer-duration ponds because, given more time to develop, they can consume a broader range of resources and achieve a larger size [50,51,54]. Carnivores are indeed found at higher frequencies––and are more extreme––in shrimp-rich, short-duration ponds [51,60]. However, in ponds that are neither highly ephemeral nor long-lasting, frequency-dependent disruptive selection—arising from intraspecific competition for food––maintains both morphs [50,59,61] (Figure 1b). Thus, because each morph is adapted to different environments that vary in time and space, selection favors plasticity and the evolutionary maintenance of both morphs (and any underlying genetic variation). 

Notably, this polyphenism is also reversible: tadpoles can switch developmental pathways from carnivore to omnivore if the prevailing environmental conditions change (e.g., a second rainfall event refills a previously shallow pond [50,54,62]). This capability provides a unique opportunity to consider both the adaptive and maladaptive morphs under pond conditions favoring one of the two morphs, as carnivores rarely develop in ponds that initially favor omnivores.

### 2.2. Natural Experiment

Our sampling focused on two ponds near Portal, Arizona, USA that contained tadpoles of *S. multiplicata*. A detailed description of the ponds and the ecological events that allowed for this experiment are described in detail elsewhere [30,48], but we describe the key features herein (depicted in Figure 2). The two studied ponds, ‘Horseshoe’ and ‘PO2-N’, are 2.8 km apart, similar ecologically, and have been breeding sites for *S. multiplicata* for over 40 years. In the summer of 2016, a single thunderstorm filled both ponds such that they were similar in size, and *S. multiplicata* bred in both ponds. Twelve days later, we observed a similar fraction of carnivores in each pond (~50% [60]). Two days after these observations, a second thunderstorm doubled the original size of PO2-N, but not Horseshoe, and converted the former from a small, ephemeral pond to a large, long-duration pond. Consequently, the density of fairy shrimp and prey tadpoles was reduced in PO2-N, and emaciated-looking carnivores were observed.

These altered conditions allowed us to decouple assessment and production. Initially, both ponds favored carnivores because they were short-duration ponds with high densities of prey for carnivores. Yet, when PO2-N transformed into a long-duration, shrimp-poor pond, tadpoles in this pond found themselves in an omnivore-favoring environment. Accordingly, some carnivores in PO2-N adjusted their phenotype via plasticity and switched back to the omnivore morph because of the reversible nature of this polyphenism. Thus, the ponds, though starting with similar conditions, ended up favoring alternative morphs (omnivores in PO2-N and carnivores in Horseshoe [48]), thereby allowing us to examine each of the assessment and response aspects of the polyphenism.

### 2.3. Sample Collection

We sought to identify loci associated with morphs (phenotypic response) and loci associated with developing as the adaptive morph (i.e., becoming a carnivore in Horseshoe and an omnivore in PO2-N versus an omnivore in Horseshoe and a carnivore in PO2-N) in each population (environmental assessment). Twenty-four days after the initial rain event (22 days after tadpole hatching), we sampled ~100 carnivores and omnivores from Horseshoe, as described previously [30]. We transported these tadpoles to the Southwestern Research Station. We reared them in wading pools, separated by morph, for three days before shipping them to our laboratory at the University of North Carolina.

Four days after sampling Horseshoe, we sampled PO2-N in the same way. The tadpoles in the latter developed more slowly, presumably because of less-dense food following the second filling. Although there were likely omnivores that never expressed plasticity in PO2-N (i.e., never became carnivores in the first place), our field observations of sampled individuals suggest that many did. Specifically, many sampled omnivores possessed omnivore-like jaw musculature but *carnivore*-like gut length, which is diagnostic of an individual that had switched from carnivore to omnivore because the gut does not revert as readily as jaw muscles [50,62]. Although the possible inclusion of some individuals that potentially did not express plasticity could increase the noise in the PO2-N omnivore samples, assignment to the other three categories was straightforward and less noisy. 

At the University of North Carolina, tadpoles and metamorphs were kept separate by population and morph but were treated the same otherwise [48]. Upon reaching metamorphosis, we took a toe clip from the individual, placed the toe clip in a labeled vile filled with ethanol, and stored the vial in a freezer kept at −80 °C. Our final sample sizes for sequencing and analysis (described below) were: Horseshoe carnivores (C_Horse_; *n* = 47) and omnivores (O_Horse_; *n* = 50) and PO2-N carnivores (C_PO2-N_; *n* = 23) and omnivores (O_PO2-N_; *n* = 20). We acknowledge that we had no replication of each pond type. This is not a problem for our design because we were primarily interested in establishing whether different loci were associated with (i) morph production (carnivores versus omnivores) and (ii) environmental assessment (those individuals that made the overall adaptive choice in which morph to become) rather than exploring a possible association between a *particular* pond type (i.e., environment) and a specific set of candidate loci.

All the procedures were carried out in accordance with all the relevant ethical regulations. Our study protocol was approved by the University of North Carolina Institutional Animal Care and Use Committee (IACUC IDs 14-297.0 and 17-055.0). Field collections were completed under scientific collection permit SP745794 issued by the Arizona Game and Fish Department.

### 2.4. Hi-C Library Preparation, Pseudogenome Assembly, and Annotation Transfer

To examine genetic variants associated with morph or adaptive choice in a genomic context, we first attempted to improve the contiguity of the recently published scaffolded assembly of the *S. multiplicata* genome [63]. We utilized a Hi-C chromatin conformation capture sequencing approach coupled with the existing genome assembly. A Hi-C sequencing library from the liver tissue of a single *S. multiplicata* individual from the same population as the published genome assembly (‘410 Pond’ [63]) was prepared using the Phase Genomics Hi-C Animal Kit (Phase Genomics, Seattle, WA, USA). The library was sent to Novogene for library QC and sequencing. We generated paired-end 150 bp reads using an Illumina HiSeq 4000 sequencer. 

We aligned the Hi-C sequence data to the previously published genome assembly [63] with ‘Juicer’ [64]. We used this alignment to generate a candidate chromosome-length genome assembly using ‘3D-DNA’ (settings: -i 4000 --editor-saturation-centile 60 --editor-coarse-resolution 100,000 --polisher-coarse-resolution 50,000 [65,66]). Because of a relatively weak signal along the diagonal in the Hi-C data, we opted to align the output of ‘3D-DNA’—without any manual adjustment—to a recently generated high-quality genome assembly of the congener, *S. bombifrons* (RefSeq assembly: GCA_027358695.2 [67]). To create this pseudogenome, we used ‘RagTag’ v2.1.0 with inferred gap sizes [68]. For downstream analyses, contigs that we did not align to the 13 chromosomes in the *S. bombifrons* genome (which correspond to the expected number of chromosomes by karyotype, as demonstrated in [69]) were excluded from the assembly used in these analyses.

We also transferred the existing gene annotations from the published assembly to the pseudogenome assembly. We utilized ‘flo’ [70] to generate a chain file as input for ‘liftOver’ [71] to transfer the annotations to coordinates within the new assembly. Additionally, we aligned the recently generated whole tadpole transcriptome [72] to the pseudogenome assembly using ‘GMAP’ version 2019-05-12 [73].

### 2.5. DNA Extraction and ddRAD-seq Library Construction

Genetic samples were extracted and sequenced as described previously [30]. In brief, genomic DNA was extracted from juvenile toad toe clips using Qiagen DNeasy Blood & Tissue Kits with a modified protocol. The DNA samples then underwent double-digest RAD sequencing (ddRAD-seq) library preparation using the enzymes SphI and MluCI following previously published protocols [74,75]. Samples were multiplexed across four libraries with 200 ng of template DNA per sample, which then underwent single-end 100 bp sequencing on two HiSeq 4000 lanes (two libraries per lane) at the University of Oregon’s GC3F facility. These data were utilized in a previous study [30].

### 2.6. Quality Control and SNP Detection

We trimmed ddRAD-seq library sequence data with ‘trim_galore’ [76], using a minimum quality score of 30 to improve alignment to the *S. multiplicata* pseudogenome. The trimmed reads were aligned to the reference genome with ‘bwa mem’ using default parameters [77], and the aligned reads were processed with the ‘samtools’ suite of command line programs [78]. Variant calling was performed using the GATK pipeline, utilizing the ‘HaplotypeCaller’, ‘GenomicsDBImport’, and ‘GenotypeGVCFs’ [79] for each population (Horseshoe and PO2-N) and both populations combined. The resulting variants were converted to bed/bim/fam format using ‘PLINK v1.9’ with settings –maf 0.05 –geno 0.5 for each population and both populations combined [80].

### 2.7. Genome-Wide Association Scans

We performed four scans to identify loci with each trait of interest (hereafter, such loci are referred to as ‘QTLs’) using FaST-LMM (which utilizes the input SNP data to generate a similarity matrix to account for relatedness between samples) [81]. The traits of interest of the four scans (see ‘Sample collection’ for individual sample sizes) are: (1) carnivores vs. omnivores across both ponds (with covariate ‘Pond’; 140 total samples), (2) carnivores vs. omnivores in Horseshoe only (97 total samples), (3) carnivores vs. omnivores in PO2-N only (43 total samples), and (4) environmental assessment capacity (i.e., carnivores in Horseshoe and omnivores in PO2-N vs. omnivores in Horseshoe and carnivores in PO2-N, with covariate Pond; 140 total samples) (Figure 2). A 5% significance threshold was used for each scan using Bonferroni correction based on the number of markers in each scan: (185,430 markers in scans 1 and 4 utilizing data from both ponds; 184,923 markers in scan 2 examining samples from Horseshoe; and 175,131 markers in scan 3 examining samples from PO2-N). 

### 2.8. Statistical Analysis of Putative QTLs

We tested for dominance at individual loci by converting heterozygous genotypes to each homozygous genotype (first major then minor alleles) and fitting both the converted and original data to logistic models using ‘glm’ in base R ([82]). We compared the result models using a likelihood ratio test in the ‘lmtest’ package ([83]) and AICc using the ‘MuMIn’ package in R [84], where any converted model with a significantly lower log-likelihood (*p* < 0.05) or ΔAICc ≤ −2 suggests that the SNP exhibits dominance. We did not examine epistatic interactions between putative QTLs because of limited sample sizes between pairs of loci.

### 2.9. Analysis of Nearby Genes

To identify genes near each putative QTL, we examined all the genes and transcripts within a 200 kb window centered on the QTL genomic position. These genes and transcripts were extracted using ‘bedtools window’ [85]. The sequence of each identified gene was compared to human, *Xenopus tropicalis*, and *Nanorana parkeri* genes using the top hit from ‘blastn’. We additionally compared all the nearby transcripts to those identified as differentially expressed between carnivores and omnivores in a previous study [72] to see whether any differentially expressed genes were in proximity to the putative QTLs.

## 3. Results

### 3.1. Improved Genome Assembly Properties

As described in Seidl et al. [63], the input draft genome had a total length of approximately 1.07 Gb. This draft consisted of 49,736 scaffolds, with a scaffold N50 of 71 kb and a maximum scaffold size of 60.2 Mb. After completion of the ‘Juicer’–‘3D-DNA’ pipeline and alignment to the *S. bombifrons* genome, our newly assembled pseudogenome has a total length of ~1.12 Gb, spread across thirteen chromosomes (Appendix A). The pseudogenome assembly includes 878.4 Mb from the original genome sequence (81.6% of the bases) and 243.5 Mb of inferred gaps, with a scaffold N50 of 129.3 Mb and a maximum scaffold size of 186.3 Mb.

Of the 42,671 putative genes previously reported [63], 38,481 (90.2%) were at least in part successfully lifted over to the new pseudogenome assembly. Of these, 10,295 gene-level annotations (24.1%) were fully placed in the new assembly, while another 28,186 genes (66.1%) had partial gene components (e.g., some exons and introns) transferred to the new assembly. Additionally, of the 19,639 genes that had previously been identified as high-quality candidates, 18,406 (93.7%) were at least partially lifted over to the new assembly.

### 3.2. SNP Detection

The GATK variant calling and joint genotyping pipeline generated 658,541 variants in the Horseshoe population, 243,131 variants in the PO2-N population, and 967,526 variants when both populations were combined. After converting to PLINK format, filtering to include only those loci present in 50% of the samples and with a minor allele frequency greater than 5%, there were 185,430 variants across the 97 samples from the Horseshoe population, 175,131 variants across the 43 samples from the PO2-N population, and 185,430 variants when all 140 samples from both populations were included.

### 3.3. Genome-Wide Association Scans

A genome-wide scan for loci associated with morph production across both ponds (n = 140) did not identify any significant peaks (Bonferroni 5% q-value = 2.70 × 10^−7^; Appendix A). Likewise, a scan for loci associated with morph production in the PO2-N population (n = 43) also did not recover any significant peaks (Bonferroni 5% q-value = 2.86 × 10^−7^; Appendix A). The ability for these scans to detect significant peaks was constrained by the small sample size and possibly mixed representation of adaptive and nonadaptive omnivores in PO2-N (‘Pond’ was included as a covariate in scans for morphs across both ponds; see Discussion). The similar scan for loci associated with morphs in the Horseshoe population (n = 97) identified five significant SNPs (Bonferroni 5% q-value = 2.70 × 10^−7^) at four peaks (defined as a set of loci within a 3 LOD score drop from the maximum LOD score, representing a 95% confidence interval; Figure 3a,c and Appendix A). These four putative QTLs are located on chromosomes 2, 3, 5, and 6 (Figure 3a, Figure 4a–e, and Table 1). For the QTLs on chromosomes 2, 3, and 5, the proportion of heterozygotes that developed as carnivores ranged from 22.9 to 38.9 percentage points higher than the proportion of major allele heterozygotes that developed as carnivores, and the proportion of minor allele homozygotes that developed as carnivores was between 39.3 and 50.0 percentage points greater than the proportion of heterozygotes that developed as carnivores (Table 1). For the QTL on chromosome 6, the proportion of heterozygotes that developed as carnivores was 68.5 percentage points lower than for major allele homozygotes. There were no minor allele homozygotes at either SNP in this QTL.

Our genome-wide scan for loci associated with the environmental assessment (i.e., carnivore morph in the Horseshoe population or omnivore morph in the PO2-N population) revealed ten SNPs (Bonferroni 5% q-value = 2.70 × 10^−7^) at nine peaks (defined as for the Horseshoe population scan above; Figure 3b,d and Appendix A). The nine putative QTLs appear on chromosomes 2, 3, 5, 7, 8, 11, and 13 (Figure 3b, Figure 4f–o, and Table 1). Across these QTLs, the proportion of heterozygotes that developed as the adaptive morph ranged from 14.5 to 66.5 percentage points higher than the proportion of major allele heterozygotes that developed as the adaptive morph. The proportion of minor allele homozygotes that developed as the adaptive morph exhibited changes from −7.4 to 54.1 percentage points from the proportion of heterozygotes that developed as the adaptive morph (Table 1). There were no minor allele homozygotes for the QTL on chromosome 11.

The environmental assessment QTL found on chromosome 2 at position 139,184,079 (Figure 4g) had ΔAICc = 2.09 for the model where heterozygotes were grouped with minor allele homozygotes, suggesting that the minor allele displays dominance at this locus. Similarly, the environmental assessment QTL found on chromosome 5 at positions 7,752,008–7,752,029 (Figure 4i–j) had ΔAICc = 2.08 for the model where heterozygotes were grouped with minor allele homozygotes. 

Two of the QTLs associated with morph production in Horseshoe and with the environmental assessment were shared between the two scans. These include the QTLs found on chromosome 3 at position 19,873,503 and chromosome 5 at position 90,293,461. Although these loci were not significant in the PO2-N population, we observed a pattern suggestive of an effect at both loci (see Discussion; Appendix A).

### 3.4. Nearby Gene Analysis

We identified a total of 90 *S. multiplicata* genes (42 of which were previously identified as high-confidence coding genes [63]) within 200 Kb windows centered around each QTL, ranging from two to seventeen genes per QTL (between zero and ten high-quality genes; Appendix A). We also identified 186 transcripts within these 200 Kb windows, ranging from three transcripts (with one isoform each) to 39 transcripts (with 52 corresponding isoforms; Appendix A). Of these, one transcript (with a single isoform) in the morph production contrast and three transcripts with five total isoforms in the environmental assessment contrast in the window around each QTL were previously identified as differentially expressed genes between carnivores and omnivores [72].

Additionally, we found a QTL contained within the body of one gene and two transcripts in each of the morph production and environmental assessment scans, including one QTL in the body of a transcript shared across both scans. The QTL located on chromosome 2 at position 9,988,823 (Appendix A), from the morph production contrast in Horseshoe, is located within an intron of the *S. multiplicata* gene g39717 (chromosome 2:9,982,401–9,995,648) and the differentially expressed (between morphs) transcript TRINITY_DN2071_c0_g1 (chromosome 2:9,982,332–9,996,359). The best BLAST hits for this gene compared to the *Xenopus* proteome are PAX3 and PAX7 binding protein 1 (*PAXBP1*). Five other *S. multiplicata* genes and 17 other transcripts were also contained within the 200 Kb window surrounding this QTL (Appendix A). The QTL located on chromosome 3 at position 19,873,503 (Appendix A), found in both the morph production and environmental assessment contrasts, was found within the body of the *S. multiplicata* gene g37552 (chromosome 3: 19,860,782–19,875,448) and an exon of the transcript TRINITY_DN6660_c1_g1 (chromosome 3:19,736,428–19,877,260), with the best BLAST hit of the ESF1 nucleolar pre-rRNA processing protein homolog (*ESF1*). An additional three genes and 24 transcripts were found in the 200 Kb window centered on this QTL (Appendix A). In the environmental assessment contrast, the QTL found on chromosome 11 at position 16,020,207 (Appendix A) is within a coding region of the gene g32894 (chromosome 11:16,019,980–16,020,880), with the best BLAST hit of the LINE-1 retrotransposable element ORF2 protein (*LORF2*). In addition, three transcripts were found within the 200 Kb window surrounding this QTL (Appendix A). Finally, the QTL located on chromosome 2 at position 77,068,752 (Appendix A) in the environmental assessment contrast sits in the exon of the transcript TRINITY_DN2576_c0_g1 (chromosome 2:77,066,799–77,070,737), which has a low-confidence best BLAST hit against several SNORD-class small nucleolar RNA (the specific SNORD that best matches varies between isoforms of the transcript). Four genes and 22 additional transcripts were also found in the 200 Kb window around this QTL (Appendix A).

## 4. Discussion

Using a newly improved genome and previously sequenced ddRAD-seq data from carnivore and omnivore tadpoles of *S. multiplicata* [30], we investigated the genetic bases of phenotypic plasticity in a novel resource polyphenism. Our results extend those from a previous study by confirming that the two components of adaptive plasticity—environmental assessment and phenotypic response [13,29,30]—do indeed have different genetic architectures.

In particular, we found five SNPs at four QTLs associated with phenotype development in Horseshoe and ten SNPs at nine QTLs associated with the ability to assess the environment accurately across both ponds. The quantile–quantile plots suggest that many other SNPs differed from expectations but were not statistically significant because of the small sample size (Figure 3c,d). Each scan identified QTLs across multiple chromosomes. These findings indicate that the morph production and environmental assessment phenotypes are both polygenic complex traits. Our results correspond with other genetic and genomic findings within this system [30], including the large number of differentially expressed genes found between carnivores and omnivores across different time points and tissues [72,86,87]. Our findings also accord with the large number of phenotypic traits that differ between the alternative morphs [51,54]—the number of coordinated traits involved in producing carnivores and omnivores suggests that many genes are likely involved in these differences.

Of the detected QTLs, two were shared between the morph and assessment scans. It seems likely that these are, in fact, environmental assessment loci. Because the environmental assessment phenotype is measured based on the morph outcome together with information about which morph is favored in each pond, it is unsurprising that we would find significant effects of morphs for these same loci within the Horseshoe population. Furthermore, when we examine the data for these loci in PO2-N, we also see a relationship between morphs and genotypes, although it does not reach the threshold for statistical significance (but see below). Thus, in total, we have identified seven QTLs significantly associated with only the environmental assessment, two QTLs that are only associated with morphs in Horseshoe, and two loci shared between the two scans, likely indicating that they are environmental assessment loci.

We found many candidate genes and transcripts in windows around the putative QTLs. Here, we focus on two candidate genes because of their positions relative to the QTLs, putative functions, and previous identification as differentially expressed genes between carnivores and omnivores [72]. First, the QTL for morph production in Horseshoe on chromosome 2 at position 9,988,823 was located near the end of an intron of the gene *PAXBP1*, which was previously found to be differentially expressed between carnivores and omnivores [72]. The PAXBP1 protein has been demonstrated to act as an adapter, linking two PAX transcription factors to a histone methyltransferase complex mediating di- and trimethylation of H3K4, regulating the expression of genes involved in the formation of skeletal muscular tissue [88,89]. This is interesting because of the significant remodeling of the jaw muscle in *S. multiplicata* carnivores. 

Second, the QTL on chromosome 3 at position 19,873,503 (found in both the environmental assessment and morph production contrasts) is found within the exon of *ESF1*, which may be involved in helping pre-rRNA assume the appropriate conformation for cleavage in the process of ribosome biogenesis [90]. This gene is intriguing for two reasons. First, most, if not all, plastic phenotypes are likely mediated through differential gene regulation [91], which may be driven by differences in ribosome composition [92]. Second, the differential expression of genes for tRNA aminoacylation (a process associated with the translation of transcripts at ribosomes) has previously been implicated in this polyphenism [72]. Although the link between these traits or the functions of candidate genes near the QTLs identified in this study in generating resource polymorphism in *S. multiplicata* is unknown at this juncture, these candidates offer an intriguing starting point for functional verification through genetic or pharmacological manipulation in future studies.

Although we caution against overinterpreting the relative number of QTLs found between the morph production and the environmental assessment phenotypes, it is crucial to consider the evolutionary implications of finding different loci associated with the environmental assessment and phenotypic response aspects of plasticity. Although more work needs to be conducted to elucidate the gene networks associated with these variants, it has been hypothesized by some that genes involved in the downstream construction of this phenotype are more likely to be labile targets of selection when compared to genes involved in upstream assessment processes because of the lack of pleiotropic restraints, which impose strong selective pressure on emergent variants at such loci involved in upstream processes [47]. At the same time, however, downstream phenotypic response genes may be under strong selection because the produced phenotypes are regularly exposed to selection. On the other hand, environmental variability in both space and time suggests that accurate assessment should be a key target of selection in *S. multiplicata*. Still, that both adaptive and maladaptive phenotypes are produced in a given environment suggests that the assessment is not perfect and that upstream environmental assessment genes may actually face relatively relaxed selection. Comparisons of our improved pseudogenome to genomes of other close relatives using tests of selection could begin to unravel these complexities.

We may expect differences between the loci involved in the two aspects of plasticity to have different signatures of selection and, therefore, different biases in detection when investigating QTLs associated with plasticity. In particular, we may expect that many of the loci involved in the phenotypic response aspect of plasticity—particularly in complex polyphenisms—exhibit little genetic polymorphism because of the strong selection the produced phenotypes experience. On the other hand, directional selection on certain loci may be mitigated by different genotypes being favored under different conditions, leaving stable polymorphic loci in the population. We also can predict that those polymorphic loci that are present are more likely to be loci of small effect sizes because of the large number of individual component traits that undergo modification in complex polyphenisms.

Conversely, it is reasonable to think that there will be more genetic polymorphism for the upstream elements involved in the initial environmental assessment, which direct phenotype production, because selection acts on such loci more indirectly. Additionally, because of the upstream nature of such loci, we may expect there to be relatively fewer of them but exhibiting large effect sizes.

The results of such biases suggest we would detect a number of loci from small to moderate effect sizes associated with phenotype production and just a few associated with the environmental assessment. However, because of the power constraints of the current study, our results accord well with these predictions. We detected more environmental assessment loci than morph production loci because, with our sample sizes, we only have the power to detect the loci with the largest effect sizes. This result corresponds to the expectation that the environmental assessment involves fewer loci of larger effects. These findings suggest that studies of genetic variation underlying phenotypic plasticity may be biased toward detecting environmental assessment loci as they are likely to exhibit these larger effect sizes. The possibility of such a bias underscores the importance of large sample sizes to get a more detailed picture of the genetic underpinnings of plasticity, especially when studying natural populations with unknown amounts of population structure and noisy environmental cues.

This work serves as a springboard for further study of plasticity’s genomic basis in this system. Such additional work is warranted for at least two reasons. First, as noted in the Materials and Methods section, our ability to detect significant QTLs in PO2-N, already limited by the small sample size, was likely further complicated by the fact that for an individual to have the adaptive omnivore morph in this population, these individuals would have had to initially become carnivores and subsequently revert to the omnivore form when the pond was refilled. However, some of the omnivores we collected from this pond might have been omnivores the entire time and, thus, did not produce the adaptive phenotype initially. Therefore, our findings are conservative because both non-plastic and adaptively plastic omnivores may be present in the phenotype data. Examining the data for the two overlapping loci between the Horseshoe morph production and the environmental assessment scans may support this view because the homozygotes that are more strongly associated with the adaptive phenotype (omnivores in PO2-N) at each locus show a relatively low percentage of omnivore production, similar to heterozygotes (Appendix A) despite no evidence of dominance effects at these loci in the environmental assessment scan or the morph production scan in Horseshoe. Thus, future research must take an experimental approach that accounts explicitly for these confounded phenotypes to separate these effects better.

Second, these data only derive from two ponds. Prior analysis of these data found little evidence of population structure between these two ponds, indicating that gene flow is occurring [30]. This is unsurprising, given the relative proximity of these two ponds. Because of this, it is unclear whether the variants found in this study are shared more broadly in the species or if they are unique to these and other nearby populations. Investigating this is an important area for further research, as tadpoles are sampled from a wider geographic range to test for the putative QTLs identified here. Doing so will also allow us to see if these loci are associated with the discrete quantitative traits that vary in the carnivore–omnivore phenotypic space [51,54].

In sum, we have identified several genetic variants associated with the two core aspects of adaptive plasticity: the abilities to accurately assess environmental conditions and generate an adaptive phenotypic response to environmental cues. These findings highlight the complex problems in unraveling plasticity’s genetic basis and serve as a foundation for further investigation into how organisms interpret and respond to environmental variability during development. Importantly, our results provide insight into the genetics of adaptive plasticity in organisms in the wild facing ecologically relevant environmental variation. These findings support the expectations that genetic variation underlies both the environmental assessment and phenotypic response aspects of plasticity and that the loci associated with these two processes are not the same.

## Figures and Tables

**Figure 1 genes-15-00611-f001:**
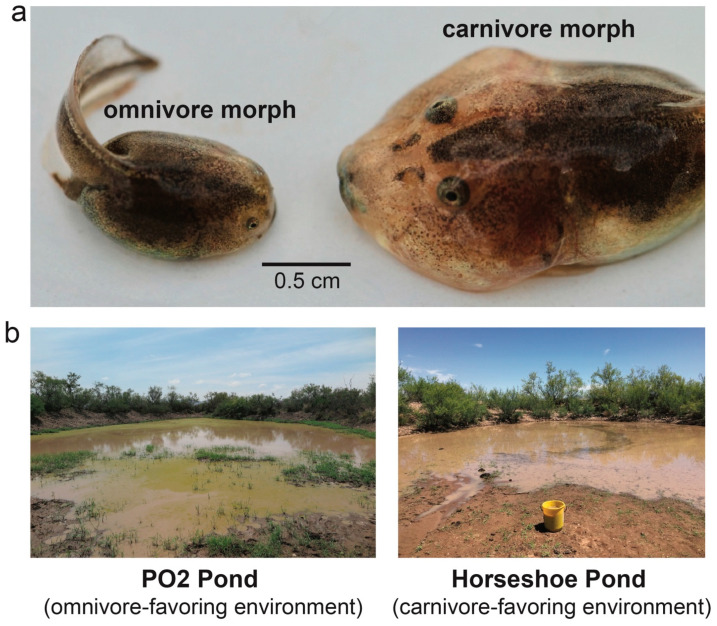
Phenotypic plasticity in spadefoot toad tadpoles. (**a**) Depending on the environment they experience early in life, the tadpoles of Mexican spadefoot toads, *Spea multiplicata*, develop into either a slow-developing omnivore morph (**left**) or a fast-developing carnivore morph (**right**). The two tadpoles pictured here are the same age (10 days old) and from the same pond. (**b**) The adults breed in temporary, rain-filled ponds that vary in duration and resources and, therefore, in which morph is favored by selection. We sampled tadpoles from two ponds: PO2 Pond, a pond in which omnivores were favored, and Horseshoe Pond, a pond in which carnivores were favored.

**Figure 2 genes-15-00611-f002:**
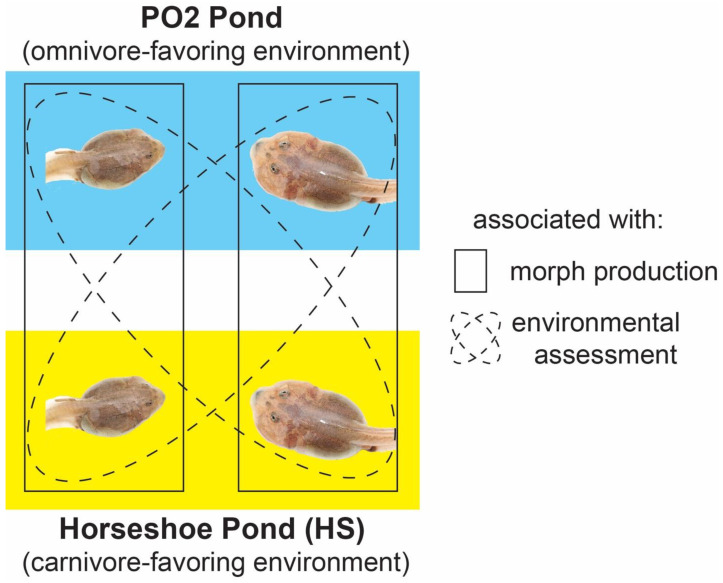
Study design. We sought to identify candidate loci differentially associated with morph production (i.e., carnivores versus omnivores) and environmental assessment (i.e., adaptive versus maladaptive morphs, depending on the pond).

**Figure 3 genes-15-00611-f003:**
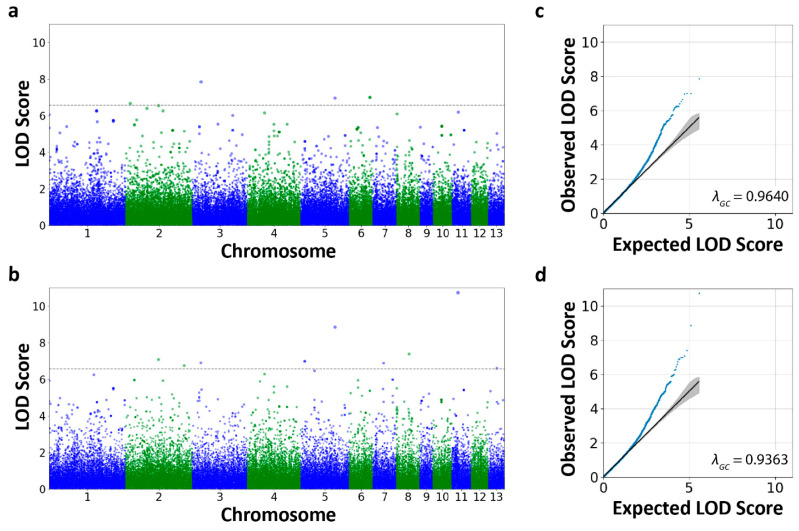
Manhattan plots portraying putative QTLs for (**a**) morph production in Horseshoe (5% Bonferroni q-value = 2.703 × 10^−7^, dashed line) and (**b**) environmental assessment (5% Bonferroni q-value = 2.696 × 10^−7^, dashed line). Quantile–quantile plots depict the deviation in observed LOD scores for all the loci relative to expected values from a null χ^2^ distribution. The solid black line represents the exact concordance of expected and observed values, so a deviation in loci to the left generally suggests a lack of power to detect true deviations from the null model. Quantile–quantile plots are shown for (**c**) morph production in Horseshoe and (**d**) environmental assessment.

**Figure 4 genes-15-00611-f004:**
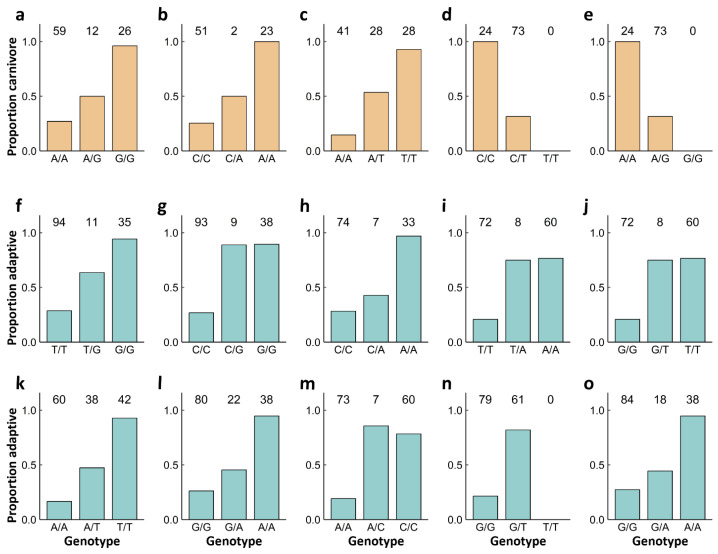
Individual association plots showing (**a**–**e**) the proportion of carnivores produced in a given genotype class at each putative QTL locus from the morph production scan in Horseshoe and (**f**–**o**) the proportion of individuals that developed as the adaptive morph in a given genotype class at each putative QTL locus from the environmental assessment scan. Orange bars denote results obtained from the morph production scan in Horseshoe; turquoise bars denote results from the environmental assessment scan. Values above each bar represent sample size for each genotype. Note that panels (**d**) and (**e**) portray adjacent loci, and panels (**i**) and (**j**) likewise portray proximal loci; each pair constitutes a single QTL peak. The QTLs portrayed here are located at (**a**) chromosome 2, position 9,988,823; (**b**) chromosome 3, position 19,873,503; (**c**) chromosome 5, position 90,293,461; (**d**) chromosome 6, position 52,066,113; (**e**) chromosome 6, position 52,066,114; (**f**) chromosome 2, position 77,068,752; (**g**) chromosome 2, position 139,184,079; (**h**) chromosome 3, position 19,873,503; (**i**) chromosome 5, position 7,752,008; (**j**) chromosome 5, position 7,752,0029; (**k**) chromosome 5, position 90,293,461; (**l**) chromosome 7, position 30,349,593; (**m**) chromosome 8, position 31,495,326; (**n**) chromosome 11, position 16,020,207; (**o**) chromosome 13, position 22,448,116.

**Table 1 genes-15-00611-t001:** All the SNPs above the Bonferroni significance threshold for each genome-wide scan, including chromosomal position, p-value, and stepwise changes from (1) major allele homozygotes to heterozygotes and (2) heterozygotes to minor allele homozygotes in the proportion of individuals developing as: the adaptive phenotype (in the environmental assessment scan) or carnivores (in the morph production scan in Horseshoe). Values of “n/a” in the final column indicate that there were no minor allele homozygotes among the sequenced individuals. SNPs in **bold** indicate those loci identified in both the environmental assessment and Horseshoe Pond morph production scans. Adjacent loci in *italics* indicate pairs of SNPs that together constitute a single QTL peak.

Scan Type	Chr	Position	*p*-Value	Percentage Point Change in Proportion between Major Allele Homozygote and Heterozygote	Percentage Point Change in Proportion between Heterozygote and Minor Allele Homozygote
Environmental Assessment	2	77,068,752	8.32 × 10^−8^	34.9	30.6
	2	139,184,079	1.81 × 10^−7^	62.0	0.6
	**3**	**19,873,503**	**1.27 × 10^−7^**	**14.5**	**54.1**
	*5*	*7,752,008*	*1.04 × 10^−7^*	*54.2*	*1.7*
	*5*	*7,752,029*	*1.04 × 10^−7^*	*54.2*	*1.7*
	**5**	**90,293,461**	**1.40 × 10^−9^**	**30.7**	**45.5**
	7	30,349,593	1.32 × 10^−7^	19.2	49.3
	8	31,495,326	4.19 × 10^−8^	66.5	−7.4
	11	16,020,207	1.84 × 10^−11^	60.4	n/a
	13	22,448,116	2.56 × 10^−7^	17.1	50.3
Morph Production (HS)	2	9,988,823	2.21 × 10^−7^	22.9	46.2
	**3**	**19,873,503**	**1.45 × 10^−8^**	**24.5**	**50.0**
	**5**	**90,293,461**	**1.11 × 10^−7^**	**38.9**	**39.3**
	*6*	*52,066,113*	*1.01 × 10^−7^*	*−68.5*	*n/a*
	*6*	*52,066,114*	*1.01 × 10^−7^*	*−68.5*	*n/a*

## Data Availability

All the raw sequence reads are available in the NCBI SRA. RAD-seq data are associated with BioProject PRJNA1098246, and Hi-C sequencing data are associated with BioProject PRJNA1098238. The pseudogenome assembly, Appendix A are openly available on figshare at https://doi.org/10.6084/m9.figshare.25572363.

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
