# Peer review of "Genetic Variants Underlying Plasticity in Natural Populations of Spadefoot Toads: Environmental Assessment versus Phenotypic Response"

_genes, 2024, doi:10.3390/genes15050611_

Round 1

Reviewer 1 Report

Comments and Suggestions for Authors

Dear Authors:

I read carefully the manuscript entitled “Genetic variants underlying plasticity in natural populations of spadefoot toads: QTL associated with environmental assessment versus phenotypic response” by Isdaner and colleagues.

The manuscript explores the genetic variants associated with the aspect of adaptative plasticity of Spea multiplicata. The experimental design with some issues and limitations is accurate and provides an important step to explore the interaction of the genome with the environmental factors under natural conditions, which is the most remarkable of the work. I suggest that the manuscript could be published as it, because it is very well written and detailed, providing an important framework to continue the study of genetic plasticity.

Reviewer 2 Report

Comments and Suggestions for Authors

The researchers designed a comprehensive study that collected both carnivore and omnivore morphs from two ponds, conducting an environmental assessment for adaptive and maladaptive traits. They then performed a genome-wide association study of phenotype and adaptive plasticity, using a "Genome-wide QTL scan". This was done after improving the scaffolded assembly of the Spea genome with Hi-C data. Four phenotype QTLs and 9 adaptive plasticity QTLs were identified. This study is intriguing and the subject matter aligns closely with the Special Issue "Genomics of Evolution and Adaptation in Animals". I have a few comments listed below:

In the methods section, specifically "Genome-wide QTL scan (GWAS)" and others, there may be some confusion regarding the difference between QTL mapping and GWAS. This needs to be clarified.

Could you specify the number of samples or data used for each group? Is this number essential for GWAS or QTL analysis? 

In the "Nearby gene analysis" section, could you provide detailed examples, including genome coordinates of detected QTLs and genes, illustrated with a figure? 

Figure legends are listed above the figures. In my review version, figures and tables are not fully displayed.
